# The Problem of Psychological Rehabilitation of Persons with Disorders of the Musculoskeletal System Acquired in Adulthood

**DOI:** 10.3390/bs9120133

**Published:** 2019-11-30

**Authors:** Tatyana Razuvaeva, Yuliya Gut, Anna Lokteva, Evgeniya Pchelkina

**Affiliations:** Department of General and Clinical psychology, Faculty of Psychology, Belgorod National Research University, 85, Pobeda Str., Belgorod 308015, Russia; gut.julya@yandex.ru (Y.G.); pchelkina@bsu.edu.ru (E.P.)

**Keywords:** disability, rehabilitation, psychological rehabilitation, lesions of the musculoskeletal system, adaptation, psychoeducation, resource of person

## Abstract

The problem of improving the quality of life of people with lesions of the musculoskeletal system is relevant to modern society. However, the circumstances of disabled people in modern Russia are characterized by the presence of many problems, including psychological ones. The aim of this study was to study the psychological characteristics of persons with acquired disorders of the musculoskeletal system and to determine the direction and content of psychological rehabilitation. In this study, we studied the characteristics of the emotional-volitional sphere of persons with musculoskeletal disorders (n = 30) acquired in adulthood, as well as scientific approaches to the study of rehabilitation and the main directions of rehabilitation of persons with musculoskeletal disorders. Clinical, psychological, and psychodiagnostic methods, alongside statistical methods of empirical data analysis, were used. It was found that persons with acquired disorders of the musculoskeletal system have a low adaptive capacity, a high level of neuropsychic stress, a low level of self-actualization and self-regulation and are not capable of the conscious planning of activities. The personal factors that intensify the manifestation of maladjustment are isolation and emotional stress when interacting with others. The article presents the main directions for rehabilitation.

## 1. Introduction

Disability is the complete or partial loss of a person’s ability to carry out self-care, move independently, navigate, communicate, control their behavior, learn, and engage in work activities [1,2].

Recently, the situation with the growth of disability has worsened throughout the world [3,4]. The suddenness of violations of the physical health of a person carries an independent traumatic effect, accompanied by a whole range of psychological changes in the emotional, behavioral, and cognitive spheres [5,6,7,8]. 

Disability acquired in adulthood is not only a problem for the disabled person, but a burden for his whole family. Families of low social status find it especially hard to tolerate disability, when disability is included in a vicious circle of the forced increase of risky forms of activity (involvement in dangerous forms of work, heavy physical labor of women and children), which lead to a greater frequency of disability in family members [9,10,11,12,13].

One of the most important and well-discussed issues in the scientific literature is the problem of integration into society of persons with acquired disorders of the musculoskeletal system, in the context of increasing their social activity [14].

Experiencing an acquired defect becomes an additional burden on a person’s mental state, leaving an imprint on the person, contributing to a possible change in self-esteem, self-attitude, emotional background and social relationships, and most often leads to the development of severe depression [15].

Medical researchers claim that that the life of a person with acquired disorders of the musculoskeletal system is changing dramatically. Breaking down the old social roles and social ties, the capabilities of the individual do not always meet the expectations of others. From the point of view of socio-psychological maladjustment, these people represent a “risk group”, the main causes of which are, on the one hand, illness, and on the other, a special social status [16,17,18].

It can be stated that, at present, the process of the socio-psychological rehabilitation of people with disabilities is fundamental to the socialization of the individual and ensures the social development of society as a whole [19,20].

Psychologists note that the effectiveness of the social integration of persons with disorders of the musculoskeletal system depends on a number of factors that contribute to/hinder the consolidation of medical, psychological and social support for this group [21,22,23,24,25].

Thus, the accompaniment organized in this area should include a system of professional activities by various specialists which creates the conditions for the subject to make optimal decisions for personal development, successful learning and socialization.

However, in the rehabilitation practice of persons with acquired disabilities, there is a contradiction between the tasks of rehabilitation and the lack of a mechanism for solving these tasks in the form of a developed concept of introducing an integrated model of social and psychological support for people with disabilities.

The analogy of the proposed complex psychological approach to the rehabilitation of disabled people with acquired disorders of the musculoskeletal system is the approach of orientation to the rehabilitation diagnosis, which is used in medicine [26,27].

On the basis of this approach, specific programs of rehabilitation treatment will be built in the future. Its essence is not only a comprehensive analysis of the general state, but also the use of data forecast dynamics of important factors. The disabled person is taught to adapt to the predictive level of their capabilities. This takes into account the psychosocial aspects of these opportunities. Thus, in addition to medical disorders, when psychoemotional disorders are corrected in a disabled person, the task is to increase his independence in performing self-service, movement, feasible domestic labor and labor activity, as well as creatively active recreation and hobbies. In our work, the psychological side of rehabilitation is justified, the general logic of which is close to the construction of a rehabilitation diagnosis. 

Psychological rehabilitation cannot only concern individuals with disabilities; it is necessary to take into account psychosocial aspects, namely the state of the family, its economic level, its cultural and gender aspects, and so on [28].

In this article, we propose an empirical study of the emotional-volitional sphere of a person with acquired disorders of the musculoskeletal system, as well as a detailed discussion of their integrated social and psychological support, as well as the correction of stereotypical ideas about disability and the formation of a new social image of people with disabilities.

We also give some recommendations for optimizing the healthcare system and the social protection of the population in the field of overcoming the consequences of acquired disability and successful integration of people with disabilities into society.

The purpose of this study was to study the psychological characteristics of persons with acquired disorders of the musculoskeletal system to determine the direction and content of psychological rehabilitation.

**Hypotheses:** 
*Disabled people with disorders of the musculoskeletal system acquired in adulthood differ in psychological status from healthy people in cognitive, emotional and behavioral areas, and are characterized by the following psychological features:*

*social adaptation and regulation of behavior has a low level of severity in combination with high nervousness (including asthenic type), irritability, emotional instability, and psychosomatic disorders;*

*sociability is reduced in comparison with healthy people, which is manifested in increased isolation and increased attention to their own gloomy thoughts;*

*self-regulation is manifested in a decrease in the ability to plan and evaluate their results, to model, to program, and to be flexible and independent;*

*the level of self-actualization is reduced, which is manifested in the desire to remember the past, lack of need for knowledge, autonomy, and self-understanding;*

*reduced severity of hardiness in all indicators: involvement, control, and risk-taking.*



The main objectives of the study are: 1) to perform interdisciplinary research of the psychological characteristics and rehabilitation of persons with disorders of the musculoskeletal system acquired in adulthood; 2) to identify the psychological characteristics of disabled persons with impaired locomotion, and to conduct qualitative and quantitative analysis of these features.

## 2. Materials and Methods

The study was carried out using the following methods: organizational-comparative methods, empirical methods, clinical-psychological methods, methods of qualitative and comparative analysis, interpretation methods, and methods of statistical data processing: Student’s t-test and Pearson’s linear correlation coefficient.

Diagnostic tools included the following methods: the questionnaire “Adaptability” (A.G. Maklakov, S.V. Chermyanin), the Freiburg personal questionnaire (modified by A.A. Krylova, T.I. Ronginskaya), “The style of self-regulation of behavior” (In .V.I. Morosanova), “Diagnosis of Self-Actualization of Personality” (SAMOAL) (A.V. Dazukina, N.F. Kalin), “Hardiness Survey” (S. Maddi, adapted by D.A. Leontiev, E.I. Rasskazova), questionnaire “Overall Self-Efficiency Scale” (Schwarzer Ralf, Jerusalem Matthias ).

The study involved 30 people, all males aged 20 to 35 years with acquired disorders of the musculoskeletal system (with disabilities), including: traumatic spinal cord disease (S32, S34)—76.7%; orthopedic trauma (S70)—10.0%; polyneuritis (G61)—3.3%; consequences of osteomyelitis (M86)—3.3%; complications after surgery (T80)—6.7%. All members of this group had clinically confirmed paraplegia and impaired sensitivity of the lower extremities and pelvic organs. A total of 53.3% of the surveyed individuals had a higher education, 30% had a secondary vocational education, and 16.7% had a secondary education. At the time of the survey, all subjects were unemployed, 40% of the respondents were married, 10% were divorced, and 33% lived with their parents. A total of 49% of the subjects had children. All of the respondents denied the presence of somatic disorders. The control sample included 30 male subjects who did not have limited health abilities (without disabilities), aged from 20 to 35 years. All subjects lived in the territory of the Belgorod region.

In order to improve the reliability of the data obtained, various types and forms of work were used with each of the 60 subjects: individual and group interviews, questionnaires, psychodiagnostics, and observation.

All of the participants provided written informed consent. The Ethical Committee Russian Psychological Society approved the study protocol.

## 3. Results

### 3.1. Adaptability

The process of comprehensive rehabilitation involves the study of the rehabilitation potential of the individual (the system of personal characteristics that allow you to actively and effectively participate in the rehabilitation process) to determine the direction of psychological assistance.

For greater clarity, let us present all of the statistically significant differences in the indices of both samples in one table.

At the first stage of the study, the adaptive capabilities of the subjects were studied. The results of the comparative analysis of the average indicators in subjects with disorders of the musculoskeletal system reveal a low level of communicative potential (2.2) and neuropsychological stability (2.5), as well as an average level of moral normativity (5.8). In subjects without disorders of the musculoskeletal system, there was revealed a high level of communicative potential (7.2) and neuropsychiatric stability (7.5) and an average level of moral normativity (5.8) (see Table 1).

Thus, the subjects with disabilities have low adaptive abilities (2.8). This means that the process of adaptation to new life conditions is difficult for them. There may be neuropsychiatric breakdowns, long-term functional disorders, conflict, and a reduced level of neuropsychic stability. In this connection, the deterioration of self-esteem and adequate perceptions of reality leads to problems in building contacts with others.

### 3.2. Personality Traits

Next, we turn to the results of the diagnosis of the states and properties of the individual, which are of paramount importance for the process of social adaptation and the regulation of behavior. In the sample of persons with disabilities, in contrast to persons without disorders, an increased level of neuroticism was revealed (7.9 and 3.5, respectively), which indicates a sufficiently pronounced neurotic syndrome of asthenic type with significant psychosomatic disorders. It is also worth noting that there is an average with a tendency towards a low level of sociability and communication needs (4.5), and an average level of openness and desire for trust and frank interactions with others (4.8). In the sample of subjects without violations, these indicators are above the average level (6.6 and 7.1, respectively) (see Table 1).

### 3.3. Self-regulation and Self-actualization.

During the study of the style of self-regulation of behavior in subjects with disabilities, the average level of planning, modeling, programming, evaluation of results and flexibility was revealed, but an extremely low result was shown on the scale of independence (1.5), which may indicate the weak development of the regulatory autonomy of the personality in the subjects of this sample. Such results may indicate the presence of a dependence on the opinion and assessment of others and the presence of self-regulation failures; in the sample of subjects without disabilities, the listed scales have high self-regulation indicators (*p* ≤ 0.01) (see Table 1). This indicates less autonomy and less developed ability in subjects with disabilities to consciously plan their activities to achieve their goals.

The obtained results were confirmed by the analysis of the data of the self-actualization of the subjects. In the subjects with disabilities, an average bordering on low level of “autonomy” and “need for knowledge” was revealed. The data obtained indicate a slight need for knowledge and self-actualization, as well as the absence of pronounced interests and aspirations in the subjects with a violation of the musculoskeletal system in contrast to the subjects without violations, in which these indicators are significantly higher (*p* ≤ 0.01).

### 3.4. Hardiness

The study of vitality revealed differences in the indicators of the engagement and resilience scales. Higher rates on these scales were observed in the subjects without violations (*p* ≤ 0.01). This suggests that the subjects with disabilities did not develop coping strategies that prevent the emergence of internal stress in stressful situations.

The low indices of resilience and involvement revealed in subjects with disorders of the musculoskeletal system suggest that in stressful situations they experience tension due to the lack of an ability to persistently cope with stress, as well as a feeling of rejection. At the same time, they are confident that the struggle and risk may affect the outcome of what is happening in life, despite the fact that success is not guaranteed.

Thus, subjects with acquired disorders of the musculoskeletal system differ from those tested without violations by having the following personality features:(1)a high level of neurotic personality;(2)an average with a tendency towards a low level of sociability and contact;(3)low level of autonomy;(4)low level of personal adaptive potential.

In order to verify the results, a statistical analysis of the data was carried out using the r-Pearson correlation coefficient, which allowed us to detect the presence and strength of a linear relationship between the values.

The statistical analysis showed the presence of a strongly significant two-way communication (r_xy_ > 0.7) between the levels of sociability and independence (r_xy_ = 0.93) and between the levels of sociability and autonomy (r_xy_ = 0.91), as well as the negative relationship between the levels of neurotization and communicative needs (rxy = −0.83), neuropsychiatric stability (rxy = −0.839) and adaptive abilities (rxy = −0.701). 

Thus, there are significant direct relationships between the results of the indicators of independence and autonomy, sociability and contact, as well as the inverse between the level of neurotization and the components of adaptability. In the development of psycho-corrective programs, we took this into account to identify the relationships between personal characteristics to enhance the results of correction.

## 4. Discussion

In the study of persons with disabilities, we identified the following features: the predominance of a high level of neurosis, a low level of self-reliance, self-control, and sociability, the presence of a fear of rejection, and unformed personal adaptational potential. There is a high level of anxiety, which is an indicator of the violations in the emotional sphere. In addition to the psychological explanation, high levels of anxiety can be caused by biological factors. 

Anxiety has been found to be common in patients suffering from rheumatoid arthritis, a type of musculoskeletal disorder. Anxiety levels correlated with rheumatoid factor levels [29].

There are persistent characterological features such as increased dependence on the environment, inactivity, social timidity, and excessive sensitivity.

It is the emotional–volitional sphere that plays an important role in the rehabilitation and treatment of a person with musculoskeletal disorders acquired in adulthood. People with disabilities in our society need high emotional stability, will, and optimism to highlight a positive direction of self-development, aimed at preserving themselves as individuals, and to continue active life. Such a positive constructive choice is possible provided that a person has adequate self-esteem, a sense of integrity of the “I”, the presence of socio-psychological activity, and an adequate attitude to his disease.

When experiencing disorders of the musculoskeletal system, resulting from injury or disease, a person undergoes a number of changes. Such changes lead to poor quality of life scores, including in patients with oral and facial injuries and fracture injuries [30]. However, with targeted and timely psychocorrection and rehabilitation work, they can be adapted to the changed conditions of their existence [31,32,33,34]. 

Our concept of the rehabilitation of persons with disabilities is a complex process that forms the active attitude of an individual to his own health, his interest in stabilizing his physical and psychoemotional state, and helps restore a positive perception of himself, society and life in general.

Psychodiagnostics, psychoeducation and psychotherapy are the main core elements of the continuous rehabilitation process.

The effectiveness of psychological rehabilitation is determined by the following areas:(1)Work with the family. The effectiveness of work aimed at the social integration of persons with lesions of the musculoskeletal system depends on the availability of psychological support for both the family itself and the individual from the family.(2)Formation of a program of social and psychological rehabilitation and correction. This allows you to designate the goals of the social and psychological rehabilitation of the client according to the state of his psychological status and to develop a detailed plan of rehabilitation measures.The overall objectives of psychoeducation are: expanding and strengthening self-regulation and self-control skills; the disclosure and realization of internal potential and abilities; the expansion and strengthening of communication skills; overcoming negative emotional states; to increase of frustration tolerance; to increase the adaptive capacity of the individual; the definition of life goals and interests; the formation of an active orientation of the person; the formation of a life perspective.(3)Psychological diagnosis and examination. This helps to identify the degree of the mental disorders and the psychological characteristics of the individual, as well as their rehabilitation potential and the prognosis for a rehabilitation course.As psychodiagnostic factors that determine the effectiveness of psychological rehabilitation and readaptation of persons with disorders of the musculoskeletal system, we distinguish: the subjective assessment of quality of life; social and psychological adaptation; interpersonal relationships; self-regulation and self-control; the prevailing emotional state; indicators of vitality and value-semantic sphere of the individual.Diagnostic data are used not only to build psychocorrection programs, but also to assess the effectiveness of psychological care.(4)Psychological counseling is aimed at forming an active attitude towards psychocorrectional work, forming motives for self-knowledge and self-development, reducing anxiety, increasing confidence in the possibilities of achieving positive personal changes, and expanding the sphere of awareness of motives. Diagnostic data are used not only to build psychocorrection programs, but also to assess the effectiveness of psychological care.(5)Psychoprophylaxis involves preventive psychological procedures to prevent negative emotional and behavioral manifestations.(6)Group psychological training. This assumes an active psychological interaction, reducing the consequences of psycho-traumatic situations and neuropsychic tension, the formation of personal prerequisites for adaptation to new conditions, and the elimination of secondary disorders of the communicative sphere (conflict, aggressiveness, irritability and other non-adaptive behaviors).(7)Individual psychological correction involves an active psychological impact, which is aimed at overcoming or weakening violations in the emotional state and behavior in order to ensure compliance with the requirements of the social environment and the needs of the individual with a violation of the musculoskeletal system.

Persons with acquired lesions of the musculoskeletal system, to a greater extent and much more often than individuals with congenital disorders of the musculoskeletal system, experience poor health and low levels of activity, and are characterized by a low-mood background, which certainly requires harmonization of the emotional state within the framework of psychocorrectional work [35,36].

The model of the individual program of the rehabilitation of persons with musculoskeletal disorders acquired in adulthood is based on the modular principle and includes the following modules: the health-saving module, the module of social and personal development and the module of crisis management. 

Thus, the program of psychoeducation builds resilience and the ability to cope with stress, contributing to successful self-realization and improved physical and mental health, as well as the successful adaptation to stressful situations. A sense of meaningfulness of existence is associated in human life with the presence of clear and achievable goals, with a sense of controllability of events [37,38,39,40,41].

It should be noted that this study has certain limitations. First, only a small proportion of individuals were included in the study sample; 30 people with acquired musculoskeletal disorders and 30 people without disorders. However, we looked at extensive factors, including psychological and social characteristics, lifestyle, past history, physical examinations, and clinical laboratory results from both samples in order to examine the impact of acquired disability on the subjects’ standard of living and rehabilitation potential. 

Secondly, the study was problematic. Due to the high level of emotional tension and isolation, subjects with disabilities at the beginning of the study showed no interest in interacting with the researchers. As a result, it had to be further explained that the results of the study would be used to improve the quality of life for people with musculoskeletal disorders acquired in adulthood, and that the information will be treated as confidential, and under no circumstances will the names of the subjects be disclosed.

However, the present study also has some notable benefits. First, it provides new evidence for problems (for example, the cognitive, communicative and emotional characteristics of individuals with acquired musculoskeletal disorders, the relationship between the personal qualities included in the rehabilitation potential, and the quality of life of people with disabilities) that are important for the implementation of health and social protection policies to optimize rehabilitation activities, but have been investigated by a limited number of studies. Secondly, we have identified the main direction and content of the psychological rehabilitation of persons with acquired disorders of the musculoskeletal system.

## 5. Conclusions

The results allow us to conclude that the psychological component of the rehabilitation potential of persons with disorders of the musculoskeletal system acquired in adulthood is characterized not only by the state of their emotional-volitional sphere (emotional instability, anxiety), but also by the features of their systems of relations with the social environment, which are very important for the formation of a positive attitude to activity. Therefore, one necessary component of the process of psychological rehabilitation of persons with disabilities and improving their quality of life is the formation of measures to correct violations in the mental sphere, as well as to form positive social-role attitudes, increase social activity and develop independence.

Only through psychological support it is possible to form and strengthen the social position, the formation of a system of value orientations, and ultimately the successful social integration of persons with disabilities into society.

## Figures and Tables

**Table 1 behavsci-09-00133-t001:** Comparative analysis of the averaged values of the personality characteristics of the subjects with disorders of the musculoskeletal system (n = 30) and without violations (n = 30).

Indicators	with Disabilities *	without Disabilities **	t	*p*-Value
Adaptability				
Communication needs	2.2	7.2	−16.414	≤ 0.001
Neuropsychic stability	2.5	7.5	−19.334	≤ 0.001
Adaptive Abilities	3.1	6.2	−10.691	≤ 0.001
Freiburg Personal Questionnaire				
Neurotic	7.9	3.5	17.676	≤ 0.001
Sociability	4.5	6.6	−7.968	≤ 0.001
Openness	4.8	7.1	−9.213	≤ 0.001
Self-regulation style of behavior				
Planning	5	8.6	−12.845	≤ 0.001
Independence	1.5	7.8	−29.598	≤ 0.001
General level of self-regulation	18	31	−23.334	≤ 0.001
Self-actualization of personality				
The need for knowledge	40	61	−23.901	≤ 0.001
Autonomy	42	63	19.710	≤ 0.001
Hardiness				
Involvement	22	48	−44.030	≤ 0.001
Hardiness	56	104	−49.136	≤ 0.001

Note: The table shows only statistically significant differences. * persons with disorders of the musculoskeletal system; ** persons without musculoskeletal disorders.

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
