# Peer review of "The Problem of Psychological Rehabilitation of Persons with Disorders of the Musculoskeletal System Acquired in Adulthood"

_behavsci, 2019, doi:10.3390/bs9120133_

Round 1

Reviewer 1 Report

Thank you for inviting me to review the paper on “The problem of psychological rehabilitation of persons with disorders of musculoskeletal system acquired in adulthood”. This is an important study and deserves to be published in Behavioral Sciences. I have the following recommendations.

1. Under the Introduction, the authors stated “M.R. Arpentieva notes that experiencing an acquired defect becomes an additional burden on a person’s mental state, leaves an imprint on a person, contributes to a possible change in self-esteem, self-attitude, emotional 38 background and social relationships, and most often leads to the development of severe depression [7].” Who is M.R. Arpentieva? I suggest to remove the name of this person. The statement can start with “Experiencing an acquired defect becomes an additional burden ….”

2. Under the Introduction, the authors stated “Domestic scientists assert that the life of a person with acquired disorders of the musculoskeletal system is changing dramatically”. For international journal like Behavioral Sciences, we do not use the word “domestic”. Please change “domestic scientists” to “medical researchers”.

3. Similarly, the statement on “Foreign psychologists note that the effectiveness of the social integration of persons with disorders of the musculoskeletal system depends on a number of factors that contribute to / hinder the consolidation of medical, psychological and social support for this group” should be amended. It cannot be “foreign” as this is an international journal. Just use psychologists.

4. Under the methods, the authors should state the name of musculoskeletal disorders. E.g. rheumatoid arthritis
5. Under the methods, the authors should state the statistical analysis.

6. Under the Discussion, the authors stated, “Line 151: There is a high level of anxiety, which is an indicator of violations in the emotional sphere.” This statement could contain bias towards psychological factor. It is important to state a reference which highlights biological basis of this finding in order to achieve a balanced view. Please add the following statement:

Line 152:…. indicator of violations in the emotional sphere. Besides psychological explanation, the high level of anxiety could be caused by biological factors. Anxiety was found to be common in patients suffering from rheumatoid arthritis, a type of musculoskeletal disorder. Anxiety level was correlated with levels with rheumatoid factor (Ho et al 2011) and interleukin-17 (Liu et al 2012).

References

Ho RC, Fu EH, Chua AN et al Clinical and psychosocial factors associated with depression and anxiety in Singaporean patients with rheumatoid arthritis. Int J Rheum Dis. 2011 Feb;14(1):37-47.

Liu Y, Ho RC, Mak A. The role of interleukin (IL)-17 in anxiety and depression of patients with rheumatoid arthritis. Int J Rheum Dis. 2012 Apr;15(2):183-7.

7. Under discussion, the authors said “Line 161: “When disorders of the musculoskeletal system, resulting from injury or disease, the person undergoes a number of changes.” It is very important to highlight the impact of such changes. Please add the following statement to offer more explanation.
Line 161:…. undergoes a number of changes. Such changes result in the lowest health related quality of life including those patients with oral and facial injuries and fracture injuries (Vu et al 2019). With targeted and timely psychocorrection….

Reference:
Vu HM, Dang AK, Tran TT et al (2019) Health-Related Quality of Life Profiles among Patients with Different Road Traffic Injuries in an Urban Setting of Vietnam. Int J Environ Res Public Health. 2019 Apr 24;16(8).

8. Line 162, psychocorrection is not the right term. I suggest the authors to change to psychoeducation.
9. From line 170 – 220, the authors should not use point-form. Please write complete sentences instead.
10. Please add a section on limitation. This study has several limitations including small sample size and unknown diagnosis of participants.

Author Response

Dear reviewer, we Express our appreciation and thankfulness for the constructive comments to our research! 

Response to reviewer's comments 1. (corrections are highlighted in yellow in the text).

In the introduction we replaced the text "M. R. Arpentiev notes that .""experiencing an acquired defect becomes an additional burden ...." (line 40). In the introduction, we replaced the phrase "domestic scientists" with the phrase " medical researchers" (line 43). In the introduction, we replaced the phrase "foreign psychologists" with " psychologists» (line 51) In the methods section we specify the name of musculoskeletal disorders (lines 109-112). The authors presented a statistical analysis using Student's t-test and Pearson's linear correlation coefficient (line 102). We provided a link that highlights the biological basis of the subjects ' anxiety and added a reviewer-recommended statement: in Addition to a psychological explanation, high levels of anxiety can be caused by biological factors. Anxiety has been found to be common in patients suffering from rheumatoid arthritis, a type of musculoskeletal disorder. Anxiety levels correlated with levels of rheumatoid factor (Ho et al 2011) and interleukin-17 (Liu et al 2012) (lines 194-196). In line with comment # 7, we added the following statement "such changes result in the lowest health-related quality of life, including in patients with oral and facial injuries and fracture injuries (Vu et al 2019) (lines 206-207). The term "psycho-correction" we have changed "psychoeducation" (line 212). At the end of the "Discussion" section, we indicated the limitations of the study (lines 258-267) From lines 170-220 we removed the dot form and wrote full sentences (lines 221-225).

Reviewer 2 Report

Review of the article
in the journal “Behavioral Sciences”
Authors: Tatiana Razuvaeva, Julia Gut, Anna Lokteva, Evgenia Pchelkina
Article title “The problem of psychological rehabilitation of persons with disorders of musculoskeletal system acquired in adulthood”
â„– Criteria for reviewing Yes No Note
1 Relevance of the study v The study is devoted to the actual problem of modern rehabilitation of patients with acquired disorders of the musculoskeletal system in the framework of the International Classification of Functioning, Disability and Health - ICF

2 Justification of the need for research (new phenomenon, gap in scientific knowledge) v
3 Scientific novelty, significance of the work v v The presented methods of determining the rehabilitation potential will optimize and predict the effectiveness of rehabilitation programs
4 Description of the hypothesis
5 Compliance of applied methods and research methods with the problem (relevance, validity)
6 Compliance of the title and content of materials v
7 Consistency and consistency of presentation of materials
8 Analysis of the declared problems
The insufficient number of works on this problem for 2017-2019 is analyzed
9 Statistical processing of materials
The article does not specify the methods of mathematical statistics
10 Results of the study (have the objectives of the study been achieved) v
11 The conclusions are clearly formulated, supported by the analysis of the results
12 The results make a concrete contribution to science
13 Correspondence of the title of the article to its problems
14 The style of an article v
15 The terminology is adequate to the research problem
16 The article has a logical structure
17 References and footnotes are correct v

Comments:
1. The novelty of the study is insufficiently substantiated. It is necessary to emphasize the novelty of the study.
2. In the structure of the article there is no hypothesis and research objectives. I consider it necessary to include in the structure of the article the hypothesis and research objectives
3. The article does not specify the methods of mathematical statistics. It is necessary to include in the description of statistical analysis procedures the methods of mathematical statistics used.
4. The list of references consists of 27 sources, including 4 articles from publications in international databases. It is necessary to include in the analytical part of the article at least 10 additional sources from international databases for 2017-2019.

Author Response

Dear reviewer, we Express our appreciation and thankfulness for the constructive comments to our research!

Response to reviewer's comments 2 (corrections are highlighted in blue in the text).

The authors substantiated the hypothesis in more detail and emphasized the novelty of the study (lines 83-94). Presented in the article the objectives of the study (lines 95-98). The article describes the methods of mathematical statistics-Student's t-test and Pearson's linear correlation coefficient (line 102, table 1). Expanded the list of references to 42 sources, including 10 articles from publications in international databases, for 2017-2019.
